# Thermal Cues Composed of Sequences of Pulses for Transferring Data via a Haptic Thermal Interface

**DOI:** 10.3390/bioengineering10101156

**Published:** 2023-10-02

**Authors:** Yosef Y. Shani, Simon Lineykin

**Affiliations:** Department of Mechanical Engineering & Mechatronics, Ariel University, Ariel 4077625, Israel; yosesh@gmail.com

**Keywords:** data transfer, haptic thermal interface, thermoelectric cooler, thermal cues, thermal patterns, thermal pulses, thermal icons, thermal communication

## Abstract

This research study is the preliminary phase of an effort to develop a generic data transfer method via human haptic thermal sensation (i.e., a coded language such as Morse or Braille). For the method to be effective, it must include a large variety of short, recognizable cues. Hence, we propose the concept of cues based on sequences of thermal pulses: combinations of warm and cool pulses with several levels of intensity. The objective of this study was to determine the feasibility of basing a generic data transfer method on haptic thermal cues using sequences of short pulses. The research included defining the basic characteristics of the stimuli parameters and developing practical methods for generating and measuring them. Several patterns of different sequences were designed considering the relevant data known to date and improved by implementing new insights acquired throughout the tests that were conducted. The final thermal cues presented to the participants were sensed by touch and clearly recognized. The results of this study indicate that developing this new method is feasible and that it could be applicable in various scenarios. In addition, the low impact measured on the user’s skin temperature represents an inherent advantage for future implementation. This report presents promising findings and offers insights for further investigations.

## 1. Introduction

Many activities in modern life involve the transfer of data via human–machine interfaces (HMIs), in which data are presented by a designated device and acquired by our senses. These activities can be divided into two general scenarios and, hence, two general purposes, namely, virtual/augmented reality (VR/AR)—in these cases, the presented stimuli simulate the characteristics of the environment, causing the desired perception (e.g., in gaming, military uses, and robotic or remote surgery, etc.)—and communication (where the stimuli represent predetermined messages or information). The volume and versatility of data are continuously expanding as technology advances; hence, the requirements for available channels and methods are expanding as well. The leading bioinspired approach for VR and AR applications involves combining the senses to increase immersion and, as a result, enhance the virtual experience. This approach has proven efficient for haptic communication and applications that require large haptic cue sets. For instance, the authors of [1] combined vibration elements with a lateral skin stretch component (which performs a rotation motion) and a radial squeeze element (which tightens a band wrapping the user’s arm), and showed that incorporating multiple forms of tactile stimulation improved perceptual distinguishability in comparison to only vibration signals. A subsequent evaluation of this system demonstrated its substantial success in identifying phonemes and words [2]. Additional prototypes of wearable haptic devices consisting of multiple modalities are listed and described in [3].

The haptic thermal sensory modality offers a novel dimension for transferring information, provided that the thermal cues are designed in accordance with the properties of the human sensory system. The thermal sense naturally serves as a data-transferring medium in our everyday routine. It takes part in perceiving the environment, assessing the temperatures of surrounding objects, and recognizing the materials of which they are fabricated. Intensive research has been conducted over decades to understand the physiology and psychophysical processes associated with the human thermal sense, showing a remarkably high sensitivity and resolution. Humans are surprisingly susceptible to changes in skin temperature, especially with respect to cooling [4]. We can resolve a difference of 0.02–0.07 °C in the amplitudes of two cooling pulses or 0.03–0.09 °C for warming pulses, and detect thermal stimuli when the skin temperature rises by 0.2 °C or descends by 0.11 °C (at rates above 0.1 °C/s) [5,6,7,8]. These values are highly dependent on various objective and subjective parameters, some of which relate to the participants, such as variance between people, the current state of mind, and the skin temperature baseline, while others relate to experimental limitations. However, these values provide a sense of the natural human capabilities we wish to harness to develop a communication datum.

The importance of developing a method to effectively use the human thermal sense as a data-transferring medium derives from its relative advantages over other mediums and the many potential applications it is expected to have. It is applicable as an alternative or complementary channel for various scenarios in which conventional channels such as vision, hearing, and tactile sensing are not applicable or sufficient (e.g., enhancing communication capability for the deafblind [9,10]; transferring discrete messages in silent environments such as security scenarios and libraries or, alternatively, transferring discrete messages in noisy environments such as sports stadiums and industrial factories, etc.). The use of thermal signals for this purpose is fraught with many difficulties due to the human factor on the one hand and technological factors on the other (for thorough reviews, see [5,11]). Due to the human factor, some of the major challenges are overcoming the limited number of sensations evoked by changes in skin temperature, the multiplicity of the parameters that influence human thermal sensitivity, spatial summation causing poor distinction between adjacent cues, and the thermal adaptation and thermal dynamics of the tissue affecting the human response and, hence, the sensation. The corresponding technology challenges that must be overcome are to create and control the desired thermal stimuli and monitor the response without influencing the heat transfer process itself.

Developing a method for generic data conveyance via haptic thermal sensation means using a thermal display to present encoded abstract information to be sensed and decoded by touch. A thermal display can take various shapes and forms, including portable and accessible devices such as mobile phones (stimulated by Peltier modules) [12], wearable items such as watches or bracelets [13], or clothing items fabricated from smart material with active thermal actuators and embedded devices [14,15,16] or stationary elements typically in contact with the user (for instance, a vehicle or theater seat). The most commonly used technologies in research are Peltier thermoelectric coolers/heaters (TECs). They are characterized by a fast and reliable response to electric driving current stimuli, forming a corresponding thermal signal.

Thermal cues must be knowledgeably designed to guarantee perceptual distinction between them with a high reliability. Some basic ‘thermal icons’ have been created by several research groups by varying the direction of the stimuli (warming or cooling), the magnitude of the temperature change, and the rate of the temperature change (ROC). The ‘icons’ are visual expressions of the patterns of the temperature change vs. time. These various studies have aimed to address different emphases (e.g., compatibility with mobile environments, comparing different body areas, emotional influence, and outdoor vs. indoor performance, etc.). However, their main objective was to investigate the use of haptic thermal feedback for human–computer interactions (HCI) intended for determining the reliability of these patterns in terms of recognizability. The initial experiments conducted in [17,18], using four different patterns with a 14–16 s duration, showed that the direction of the change in the temperature (warming or cooling) is the most salient parameter of a thermal icon, achieving a 97.4% identification accuracy (for individual parameter identification). The overall identification rate for all four tested thermal icons was 82.9%. The ROC was less perceptually distinguishable, even though it was combined with magnitude to create an integrated subjective intensity parameter, with an 85.4% accuracy for the individual parameter for two levels of intensity (strong: 6 °C at an ROC of 3 °C/s, moderate: 3 °C at an ROC of 1 °C/s). In [19,20], a set of six different icons was designed, each 8 seconds long, with the two optional directions (warming and cooling), one consistent magnitude of change (6 °C), and three different levels of ROC (0.7, 1.5, and 3 °C/s), showing, in addition, that presenting a dual signal yields a high distinction from the one-pulse stimuli, with a 99% accuracy. The intensity represented by the ROC remained a moderate perceptual distinguishing parameter, with an 86% accuracy for 1.5 °C/s and 82% for 0.7 °C/s.

Seeking a leading concept for an effective set of thermal cues, we observed existing binary generic-data-conveying methods, Braille and Morse codes. Effectiveness in this context means allowing more data flow per time unit. Inspired by insights from these two methods and equipped with the conclusions from the reported work on thermal cues mentioned above, we propose the concept of sequences of quick thermal pulses for a haptic data transfer method. This concept implements the advantages observed in other methods and approaches and attempts to compensate for the inherent relative disadvantages of the thermal medium, such as its poor spatial sensitivity and slow response. The idea of creating temporal sequences of stimuli was adopted from Morse code, and the confidence in the human ability to acquire a new skill that involves an extraordinary natural capability was inspired by the acceptance of the Braille code. The thermal icons evaluated to date have extended between 8 s and 16 s per signal and have offered minimal variety, with an insufficient reliability. The proposed concept applies the excellent distinguishability that has been observed between warm and cold stimuli and between single and dual stimuli. It also assumes that this capability can be expanded to a trio of stimuli. The inferior performance regarding stimulus intensity, showing only a moderate ability to distinguish between levels of temperature rate of change (ROC), is attributed to the fact that the test procedures in the reported experiments required not relative distinction, but rather absolute recognition. The proposed concept herein requires a relative distinguishability between sequential pulses of varying intensities. This fundamental difference between these two capabilities is assumed to lead to an excellent distinction between intensities as well, provided that the cues are designed appropriately. Ultimately, the short duration of cues based on a few short pulses, alongside the potentially large variety of different sequences, are basic requirements for effectiveness.

This study aims to prove the feasibility of transmitting information using haptic thermal signals that are composed of different sequences of short pulses. In practice, this means designing and creating a set of thermal-pulse-based patterns and empirically showing that they are highly distinguishable and recognizable. An integral part of this feasibility study was to knowledgeably determine criteria for designing a library of patterns, including the required properties of the pulses (mainly magnitude, duration, and the interval between pulses) and potential combinations.

After this introduction, the paper lays out a scientific background with relevant information and aspects to support the research, and presents the research methodology and the experimental results, followed by a discussion analyzing some major aspects of the findings and establishing conclusions for further research and future implementation.

## 2. Background

The haptic thermal sense is composed of several processes. This section analyzes the main aspects of these processes, showing their impact on the test methodology and on the design of the thermal cues.

The physical process that occurs in our skin in response to a thermal stimulus is conductive heat transfer. The process initiates with a heat exchange on the skin surface that leads to heat flow through the skin tissue. A thermal stimulus can be applied to the skin either by contact with an external object of a different temperature or, in a scenario more relevant to the current research, when the thermal equilibrium at an ongoing skin–object interface is disrupted by a change in temperature imposed on the object. In both cases, the heat transfer starts with a heat exchange between the skin and the object due to the temperature difference and progresses in-depth to the tissue. The heat flow intensity is determined by the thermal properties of the skin, the thermal display, and the interface. The heat within the tissue is distributed in accordance with the boundary conditions—the thermal stimulation on one end and the regulated body temperature on the other.

A heat transfer process has, by definition, two aspects: one is the energy flow in the direction from a higher temperature level towards a lower temperature level, and the other is the inevitable change in temperature along the heat transfer path. The relation between these two aspects is the heat capacity of the skin, a physical property defined as the amount of heat (measured in joules) needed to produce a temperature change of one Kelvin for a unit of matter (skin tissue). The temperature change with time due to the heat flow through the skin tissue results in a slight restraint to the process as it gradually moderates the temperature gradient that has been formed. However, since this research focuses on short stimuli, the impact is minor. However, the skin surface temperature change during the interface heat exchange is significant. The immediate response to the stimulus is a rapid temperature drop or jump (for cooling or warming, respectively), attempting to equalize the two sides of the interaction, thereby continuously reducing the gap. This continuous change in skin surface temperature has a major impact on the response to a given stimulus. Hence, it must be considered when designing thermal cues.

The temperature transient that occurs in the skin following contact with an external object is influenced by its thermal characteristics. Therefore, the human ability to perceive temperature transients makes it possible to recognize the type of material being touched. This is the idea behind material recognition based on thermal cues. The very same capability for noticing and distinguishing between stimuli is the basis for the current research. This analogy is relevant because the cutaneous reaction to a transient dynamic thermal stimulus resembles the initial stage of the reaction to contact with an external object. However, there are several differences between the two situations that must be considered when designing the thermal cues and analyzing the test results. In this research, the skin surface is in continuous contact with a controlled thermal display that presents predesigned cues that are based on sequences of pulses, and the participant is required to subjectively identify the different cues. The following are some of the major aspects that are unique to this situation vs. material recognition:The thermal recovery that occurs in the skin once the heat pulse ends is driven by the controlled temperature of the thermal display. However, in the case of extended contact with an object, the outer boundary condition, following the initial rapid temperature drop or jump, is the skin–object interface heading towards the equilibrium temperature. In this context, the pattern design must guarantee that a neutral sensation is regained between consecutive pulses to verify the temporal distinction between them.Normally, material recognition involves a combined tactile sensation, whereas our tests examine the thermal sense alone. Hence, distinguishing between pulses and recognizing patterns is expected to be more challenging, requiring a design with a more distinct differentiation.A controlled thermal display acts like a material with extremely high thermal coefficients, causing an extreme cutaneous response as the control system provides an ‘endless’ thermal capacity. The thermal display acts, in this case, more ‘aggressively’ and less tolerantly to the attempt of the skin to change its temperature, thereby causing an ongoing heat flow and, hence, a dominant thermal sensation.

The sensing mechanism involves a physiological–biochemical process that creates nervous signals in reaction to the physical process and a psychophysical process that interprets the data, establishing the final perception. Due to the structural and functional differences between warmth and coolness nervous sub-systems, the human temporal processing of dynamic thermal stimulation differs for warm and cold senses. The response time to thermal stimuli was found in [5,21] to be more rapid for cool stimuli (0.3–0.5 s for an ROC greater than 0.1 °C/s) compared to warm stimuli (0.5–0.9 s). An investigation of the physical–perceptual correspondence of dynamic thermal stimulation [22] concluded that the sense of cold is more transient than the sense of warmth, and therefore responds more rapidly to transient changes (dT/dt). When dealing with pulses that extend for only fragments of a second, the different time reactions we have to warm and cool stimuli impact our sensation, and therefore must impact the stimuli design.

Psychophysical processes influence the sensitivity thresholds and dominate the actual thermal perception. Thermal adaptation is a key phenomenon imperative to acknowledge for a basic understanding of thermal perception. Thermal adaptation occurs within the range of 30–36 °C, causing the sensation to be neutral as long as the temperature remains constant. A deviation from the action potential transmitting rate balance due to a change in skin temperature, upward or downward, evokes a warmth or coolness sensation, respectively. Beyond the neutral range, the sensation continues, even with constant temperatures. The outcome is that, within the neutral range, the thermal sensation reveals a change in skin temperature rather than the actual temperature level. In other words, our thermoreceptors are always exposed to one temperature level or another and constantly sense the local temporal temperature and react accordingly. Even so, the sensation remains neutral regardless of what that the temperature is, provided that it is constant and within the neutral range. Only a change in temperature causes a thermal sensation, as the receptor’s reaction to it is translated into a feeling of warmth or coolness depending on the direction of the temperature change. The intensity of the thermal sensation is proportional to the rate of the temperature change. A complementary phenomenon that has direct practical implications is the dependence of the thermal sensitivity threshold on the rate of the skin temperature change (ROC) [5,6,7,8]. This means that the thermal sensation is evoked by a change in temperature only from a certain minimal ROC, under which, it will remain neutral, despite the actual change that is occurring. From a different perspective, it means that, to create a noticeable separation between consecutive stimuli, it is not necessary to reach a complete cessation in heat flow and temperature change. This conclusion obviously has a great impact on the thermal cue design.

This research includes determining the characteristics of the stimuli parameters, as well as developing practical methods for generating and measuring them. Since the goal is to create a set of thermal cues that will be noticeable and distinguishable by practically everyone in maximum circumstances, a fundamental question is how to control the cues to guarantee similar impacts on all users, causing similar thermal sensations. In other words, how to control the thermal cues in a way that the temperature change or heat flux by the thermoreceptors will be indifferent to the uncontrolled parameters that normally influence the human reaction to thermal stimuli.

The thermal display temperature can intuitively seem like an appropriate measure of the intensity of the stimulus. However, the actual cue that enters the skin is delayed by the thermal resistance of the contact zone. The contact resistance causes the temperature of the skin fragment under testing to differ from the temperature of the thermal display. The extent of this delay varies and is difficult to determine because the resistance coefficient is influenced by many parameters, such as pressure, humidity, hairline, and oiliness. Hence, controlling the stimuli via the thermal display temperature lacks reliability and efficiency.

Attempting to skip the contact resistance and monitor the skin surface temperature directly encounters technical limitations that prevent the acquisition of a reliable result. The challenge is to monitor the skin fragment that is in contact with the display, but without the sensor touching the display. The sensor must be placed directly at the skin–display interface, but then it inevitably disturbs the heat exchange process between the display and the skin at the very area being monitored. Furthermore, direct contact with the display distorts the thermistor’s input; thus, the measurement does not purely reflect the skin temperature. Distancing, or, alternatively, isolating the sensor from the display adds a bypass to the heat flow path, driving it via the skin fragment from the contact zone to the sensor. This results in a reduced measurement that reflects an area of the skin that is not in direct contact with the thermal display. Due to the significant thermal resistance and heat capacity of human skin, this factor may be meaningful. Aside from these technical issues, the true skin surface temperature, even if it could be reliably measured (by neglecting or evaluating the disturbance or by avoiding it, for instance, by using a thermal camera), lacks validity as a controlling parameter, since it is influenced by the individual’s temporal thermal properties, hence impacting the temperature change sensed by the thermoreceptors for a given stimulus.

An alternative approach is to find a way to monitor the heat flux. The heat flux pumped through the surface of the thermal display is directly sensed by the thermoreceptors. This is under the assumption that the heat flux from the thermal display into the skin is unidimensional, and the losses due to the heat distribution along the skin surface are assumed to be negligible. Thus, the best thermal stimulus is the controlled heat flux. However, the heat flux is very difficult to measure; hence, it is difficult to implement its feedback control. The cutaneous temperature change detected by the human sensing system is an expression of the heat flow generated by the thermal stimulus, and so it indicates the intensity and direction of the stimulus. A warming stimulus means that the heat flow is directed into the body, and a cooling stimulus means that the heat flows outward from the body. Therefore, the heat sensation is a measure of the heat flux. The opposite is true as well: if the heat flux could be measured reliably, it would enable predicting the corresponding heat sensation.

A thermoelectric cooler (TEC) is a solid-state heat pump that employs the Peltier effect. TECs pump heat from one surface to another using electrical power. The heat flux coming into or out of the TEC surfaces is proportional to the electrical current applied, times the temperature of the surface in Kelvin. Since the variations in the thermal display’s absolute temperature are relatively small, the heat flux is almost proportional to the electrical current driving the TEC. To be more precise, the TEC full heat equation includes two terms that represent the energy waste due to the resistance of the TEC material itself, one thermal resistance and the other electrical resistance. However, the equation defines the steady state achieved with time, and at the initial stage of dynamic stimulation, these terms are negligible. The heat flow direction is in accordance with the direction of the electrical current.

A TEC is an excellent candidate to serve as a thermal display in experimental research. One side of the TEC is maintained at a predetermined, stabilized temperature, causing the positive or negative temperature difference forced by the input current to impact the other side, hence creating the desired thermal stimulus. The approach considered in this study is to control the electric current driving the TEC rather than directly controlling the thermal stimuli. In practice, the correlation between the electric current and the thermal cue displayed is not perfect due to a physical limitation we show during the experiments: although the TEC backside temperature is stabilized, the control circuit cannot react fast enough to thermal pulses, and a pulse-like change in its temperature is observed, thereby decreasing the desired temperature change on the front surface serving as the thermal display. However, this seems to be a consistent and predictable phenomenon with only a quantitative impact. Hence, the correlation is assumed to be acceptable for this study, which is focused on evaluating the concept of pulse-based patterns, particularly due to the understanding that recognizability is based on the relativity of the thermal sensation comparing adjacent pulses.

A specially designed voltage-controlled current driver, known as a transconductance amplifier (TCA), was fabricated and implemented. The stimulus is generated as a voltage versus timetable and fed to the TCA from a laboratory voltage generator. The TCA transforms the input voltage into a proportional current and drives the TEC with it.

An original modeling tool was developed in [23] and used in the current research to simulate the human response to thermal stimuli. The human skin tissue model was developed using the one-dimensional finite difference method. The thermodynamic parameters of the skin were obtained from [24,25]. On the inner side of the skin, the boundary condition is the typical human core temperature of 37 °C. The outer boundary condition is one of two alternatives; in the absence of a stimulus, it is the room air temperature, taking into consideration the thermal resistance of the convection exchange, and in the presence of a stimulus, the boundary condition is the heat flux from the thermal display. Figure 1 depicts a schematic illustration of the skin structure (a) and the finite difference model in the form of an equivalent circuit (b), where resistances represent the thermal resistance of the mesh node, the capacitors represent the heat capacity of the node, voltages are equivalent to the temperature, and currents are equivalent to the heat flux. The boundary conditions are represented as a voltage source in the temperature boundary condition case and a current source in the heat flux boundary condition case. The entire volume of tissue involved is divided into a network of elementary volumes (nodes). The minimum number of nodes in this network is chosen so that the heat accumulated by the thermal mass of the elementary volume is significantly less than the heat flowing through it [26].

The model enables the simulation of heat flow and temperature at any given location throughout the system, including inside the tissue volume, for instance, at the depth of the thermoreceptors. The model also allows us to adjust the parameters according to a variety of skin types, test conditions, or thermal display characteristics, and simulate the response to different stimuli. The thermal signals for the current study were designed with the aid of this simulator.

## 3. Methodology

The concept of thermal pulse cues was chosen, the initial pulses and patterns were designed with the aid of an original modeling system [23], and an appropriate experimental laboratory setup was fabricated. To achieve the goal of proving feasibility, the tests involved only the researchers, with no additional participants, and included repetitive trials and errors, during which, the various parameters were adjusted to find the optimal tradeoff for the minimum cue duration and maximum recognizability. The varying parameters were the rate of the skin temperature change, or equivalently, the amplitude of the stimulating current, the duration of each pulse, and the time interval between the pulses in each sequence composing a cue. The degree of pattern recognition derives from the ability to notice every pulse in a sequence and refer to each pulse individually and the ability to distinguish between varying intensities. Regarding the direction of the pulse, i.e., cooling or warming, due to practical reasons partially explained herein, the tests at this stage involved only cooling pulses. Warming pulses will be introduced in future research as separate cues and as combined cues with cooling pulses.

The tests conducted in this study are introductory for a future large-scale experiment with a statistical group of participants, during which, the final design will be determined and reliability verified. The current study is necessary for establishing the format of the thermal patterns that will be tested. Although these tests are based upon subjective thermal perception, as sensed and processed by only a few participants, we believe they were carefully conducted, allowing for a significant margin of error, thereby providing a robust prediction of the common response. In any event, the established quantitative patterns will be described with the necessary degrees of freedom to allow for further adjustments if necessary, without affecting the qualitative design. The dependent variable in these tests was the human perception and recognizability of the different stimuli, and the independent variable was the thermal stimulus displayed to the skin.

### 3.1. Apparatus and Test Layout

A Laird Thermal Systems model UT-15-200-F2-4040-TA-WG Peltier thermoelectric cooler (TEC) was used to create the thermal display. One side of the Peltier element served as the display, and the other side was mounted on a thermostatic hot-plate TE Technology CP-200HT using thermal grease to guarantee full thermal contact between them. The stabilized temperature was maintained by a TC-720 controller using a thermistor to monitor the interface temperature. Another identical thermistor was attached to the hot plate surface at some distance from the TEC, indicating the controlled temperature. The arbitrary signal generator RIGOL DG-1022 was used to form the electrical stimuli, which were delivered to the thermal display via a custom-made current driver designed to amplify the signal with a sensitivity (Iout [A]/Vin [V]) of 1.72 [A/V] (see Figure 2). An Agilent 34792a Data Acquisition System with Agilent commercial software, version 2.7, was used to process the data for analysis. The temperature change throughout the experiments was monitored at various positions using MP 3176 thermistors (5 kΩ, 0.9 mm diameter). The thermistors were mounted on the thermal display with layers of Aerogel thermal insulator to thermally isolate the zones of interest (see Figure 3). The complete stationary laboratory layout is shown in Figure 4.

.

### 3.2. Procedure

A set of seven different sequences was tested in this feasibility study—see the description in Table 1. The number of pulses per sequence was either two or three. A preliminary assumption regarding the thermal perception sensitivity to the stimulus intensity was made, claiming that the number of different intensities in each pattern must be limited to two. The evaluated pulses were designed accordingly. During the study, this assumption was slightly corrected, as it was found that, in the case of uni-directional intensification, i.e., either increasing or decreasing intensities, as in test # 6 vs. #5 or #7 in Table 1, it is feasible and even preferable to use three intensities.

The participant was first acquainted with the various sequences that were included in the study by using the visualized description and verbal definition (see Table 1). For every sequence, first, the initial values of the pulses were determined and applied to the thermal display repetitively every 8 s for a total of several minutes. The participant then rested a given hand on the display, carefully positioning the thenar eminence at the base of the thumb in a natural motion (the pressing force was not monitored), ensuring full skin contact with the thermal display and coverage of the entire area. By this act, the monitoring thermistors mounted on the display (see Figure 3) were inevitably located at their respective functional positions. This position was maintained steady until thermal equilibrium was acquired, which was determined by the readings of the interface thermistors. The steady-state temperature between the stimuli was set to approximately 32 °C. This was achieved, per pattern, by pre-adjusting the temperature of the hot plate under the back (hot) side of the Peltier TEC.

As steady-state conditions were reached, several additional cycles of the thermal cue were applied, and the participant was expected to determine what thermal sensation was perceived. The feedback included reporting answers to two questions: (1) How many pulses did you sense per sequence? (2) Which one of the optional sequences was presented? For tests #3 and #4, the feedback included an answer to an additional question: How do the two duos compare?

This procedure was repeated, first according to the build-up order of appearance in Table 1 and then randomly at four locations per participant—at the thenar eminence and the wrist of each side, while adjusting the parameters according to the feedback. As a result of each iteration, further assumptions were laid out, and the cues for the next tests were designed in light of these to fill the remaining gaps. Finally, the trio sequences—tests #5 through #7—were applied by the experimenter in a random order without notifying the participant (single-blind test) to verify unbiased recognition. The tests were also repeated on different days to increase the variance of the uncontrolled conditions that influence thermal perception, such as preliminary skin temperature and mindset. This iterative process continued as data were gathered and provided more understanding of the borders and thresholds of recognizability, and the intensities and temporal parameters were adjusted accordingly until recognition was optimized.

### 3.3. Data Analysis

The data gathered from these tests included the stimulus specification in terms of the input current and consequent voltage on the Peltier TEC, the temperature changes monitored at the different locations during exposure to the stimuli, and, finally, the subjective thermal perception reported by the participant that included the icon selection and a verbal description of the thermal sensation. All the iterations were recorded and analyzed. Due to the nature of this evaluation research, the data gathering and data analysis were conducted in parallel, with mutual feedback, in an ongoing trial and error process (see Figure 5). The experiment was completed when the stimuli temporal and intensity parameters were optimized and a full understanding was reached as to the correlation between the stimuli and thermal sensation for the various sequences.

## 4. Results

The results reported in this section include the final designs that showed convincing recognizability and the main quantitative and qualitative parameters that were learned to impact the cue design. They are interpreted herein in relation to the research objectives and described together with the experimental conclusions that can be drawn from them. The cues are presented graphically, showing the electric stimuli and the resultant display temperature and interface temperature curves. Recognizability was determined by comparing the subjective perception reported by the participants to the applied cue and could not be quantitatively measured.

Tests #1 through #4 served to determine the initial characteristics of the pulses to be used for assembling the thermal patterns for evaluation. The thermal response to the applied stimuli is presented in Figure 6. These tests showed that two consecutive cooling pulses of different intensities, applied to the thenar eminence at the base of the thumb or to the wrist, were well noticed and clearly distinguishable (i.e., one could determine which of the pulses was of a higher intensity and which was lower) when they complied with the following values:Minimal pulse duration: 250 ms.Minimal ROC at the skin–display interface: 1.3 °C/s. Corresponds to an input current of 5A when using the existing test setup.Intensity ratio between adjacent different pulses (due to duration, input current, or a combination of the two): 1.5.

The two pairs of pulses applied in test #4 were sensed as identical; that is, they were both sensed as a sequence of a low-intensity pulse followed by a high-intensity pulse. Even though the high-intensity pulses in the two cases were totally different, one had a longer duration and a lower current, and the other had a shorter duration and a higher current. The difference between the pulses is depicted in Figure 6 by the display temperature curve (the slope, representing the temperature change rate, is steeper in test #1 than in test #2, corresponding to the different currents) and is more visually obvious in the electric stimulus curves. This result shows that we cannot discriminate between two pulses of similar intensities but different designs when they are in separate sequences. The reference to pulse intensity at this stage is in terms of the product of the stimulus current amplitude and duration, the two factors that determine the amplitude of the skin temperature change, and the thermal energy delivered to or from the skin (later in this section, the adjacency of the pulses in a sequence will be added as a factor that impacts the intensity perception). This human limitation grants flexibility in designing thermal cues, as each cue must be coherent only within itself, allowing for the optimal differentiation between pulses; however, matching the pulse designs between cues is not required.


Figure 6Temperature change vs. time in response to Duo-stimuli. The stimuli are presented as the electric current applied to the TEC (brown, dotted), the resultant thermal stimulus displayed by the TEC as monitored on the display with isolation from the skin (blue, dashed), and the cutaneous thermal response as monitored at the thermal display–skin interface (green, solid). The graph randomly captures one of the many repetitions of the respective sequence applied periodically every 5 seconds. (**a**) Results from test #1: A sequence of two pulses with different intensities due to stimuli current—1st of 5.1 A, 2nd of 7.6 A. (**b**) Results from test #2: A pair of successive pulses with different intensities due to stimuli duration—1st of 250 ms, 2nd of 400 ms.
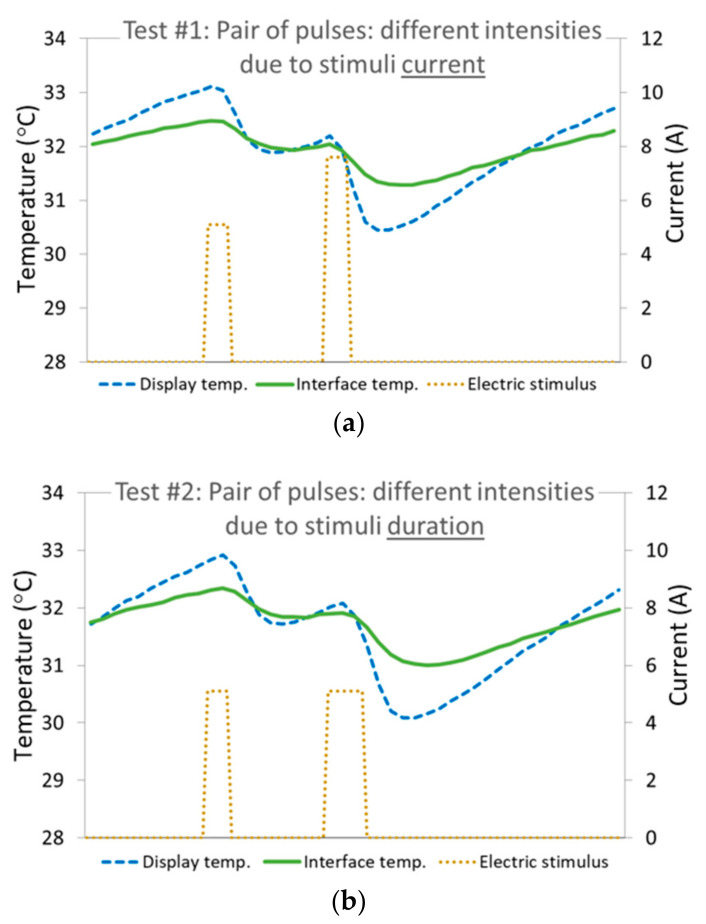



The final designs reached for tests #5 through #7 that were found to deliver the desired thermal sensations are presented in Figure 7 and Table 2. The tests showed that the adjacency of pulses impacted the thermal sensation, forcing variations in the pulse intensities to reach a desired sensation. It was found that the order of appearance in the sequence influenced the sensed intensity of a given pulse; the later it appeared, the more intense it felt. This phenomenon is manifested well by the comparison of the trio cues as they were sensed vs. the actual stimuli applied (see Table 2). Several emphasizing examples: in test #5, the second pulse was intensified relative to the first pulse; therefore, a slight difference in favor of the second pulse (9.8 A vs. 8.4 A) was sufficient to create the desired significantly distinguishable difference. The similar second and third pulses in test #6 created the desired sensation of an increasing intensity. The first pulse on test #7 required an extra boost for it to be clearly sensed as a high-intensity pulse; therefore, the duration was extended to 300 ms, and so forth.

A summary of the parameters that were found to influence the intensity of the thermal perception:The rate of skin temperature change (ROC) was determined by the amplitude of the stimulus current.The temporal duration of the stimulus current (which was multiplied by the ROC, determined the amplitude of the skin temperature change).The adjacency to a previous pulse intensified the sensation (for unidirectional stimuli, as tested in this study).

The results show the high feasibility of a generic-data-conveying method based on thermal haptic sensation by using combinations of up to three pulses. They demonstrate recognizable cues with a duration of up to 2.75 s, composed of sequences of 250 ms cooling pulses, with intervals of 1 s between them. The tested cues had a low impact on the skin temperature. The temperature drop per pulse was usually a fraction of a degree Celsius starting from approximately 0.2 °C, and the total drop for a trio cue was less than 2 °C. The tested patterns are not necessarily the final design ready for implementation, and they do not cover the full variety of pulse cues that can potentially be developed. Their role was solely to prove the concept. Further work is necessary to design a full set of cues and determine their reliability.

## 5. Discussion

The pulse cues were found to have inherent advantages and features that reinforced the robustness of the concept, other than the obvious advantages that encouraged choosing this concept: a short duration and a potential variety of combinations. In this section, we discuss some eminent aspects from the perspectives of previous studies that have evaluated thermal cues and of the study objective.

### 5.1. Preliminary Note: The Decision to Include Only Cooling Pulses

As long as the skin temperature was maintained within the neutral thermal zone (approx. 30–36 °C), it was not the nominal skin temperature that determined the thermal sensation, but rather the temperature change. This known physiological fact is demonstrated by the graphs in Figure 7, as the relative intensities of the sensations (indicated by the respective caption) were totally non-correlative to the measured temperature. The graphs also show a temporal delay of approximately 200 ms between the stimulus onset and the thermal cutaneous response. We expect a longer delay for warm pulses due to nervous differences [22]; hence, combining cooling and warming pulses requires special synchronization, adding complexity that is unnecessary at this time. This is the main reason it was decided to include only cooling stimuli in this phase of the study.

### 5.2. Cue Design: Pulse Intensity and Duration, and Intervals between Pulses

The experiments showed that raising the stimulus electrical current to improve noticeability helped only up to a certain extent. The high–low intensity ratio must remain as determined empirically, at approx. 1.5; otherwise, the low-intensity pulse is at risk of being unnoticed, and therefore, it must be raised as well. However, since pattern recognition for sequences of pulses is based on relative sensation rather than absolute sensation, recognizability did not improve significantly once the pulses were intensified enough to be noticed with a high reliability. Therefore, the working point determined as the optimal combination was close to the minimal intensity for practical and safety purposes for the experimental phase, as well as for future applicability.

A major factor of the research goal was to aim for a minimal duration of thermal cues as a basic condition for an effective data-transferring method. Therefore, the values of the pulses designed in these experiments refer to the optimal thermal cues in terms of recognizability vs. temporal shortness. Longer pulses would have been even more noticeable and would perhaps have led to improved recognizability, but the tradeoff limited the duration. Another advantage of minimal duration is that it linearly decreases the thermal impact on the skin as well, thereby minimizing the recovery time between cues. The optimal interval between pulses, with the existing experimental layout, was found to be 1 s. With short intervals, the skin’s thermal mass stalled its response to the displayed cue, thereby negating the remains of the intervals and smoothing the curve. A future investigation based on broad experiments may allow for moving the working point towards a shorter duration with or without intensifying the pulses. Furthermore, changing the spatial dimensions of the interface by enlarging the thermal display area or adding displays (implementation scenario dependent) would improve sensitivity, thermal perception, and most likely recognizability, therefore maybe allowing for a shorter duration.

### 5.3. Temporal Sensitivity: Pulse Cues vs. Extended Cues

The mechanism used to discriminate between thermal cues was different for the two types of cues: sequences of short pulses as opposed to extended pulses. As mentioned, for the sequences of short pulses, recognizability was based on the ability to discriminate between the adjacent stimuli in the sequence. A major factor causing the difference was the characteristics of human thermal temporal sensitivity.

Test #4 showed (by comparing the two graphs in Figure 6 that show the skin responses to each of the duos separately) that, for short pulses, we cannot discriminate between two cooling pulses of similar intensities. This means that we do not sense the absolute ROC, but rather discriminate between two different ROCs if they have the same duration, resulting in different intensities or total temperature changes. Conversely, we do not sense the absolute duration, but rather discriminate between two pulses of different durations if they have the same ROC. Specifically, in the discussed cases, one cannot distinguish between events of 250 ms and 400 ms solely by sensing a difference in duration, since they are both perceived as simply ‘short’, with blurred temporal starting and ending points. For extended waves, on the other hand, the temporal sensitivity becomes more sufficient. In fact, [19,20] evaluated several thermal patterns of 8 s durations, showing over an 80% accuracy in the perceptual discrimination between step- and ramp-type waves (referring to the skin thermal response), although both had the same total temperature change. The step started with a relatively short pulse causing an ROC of 1.5 °C/s at the skin–display interface, followed by a steady temperature, and the ramp had an ROC of 0.7 °C/s with a double duration. In addition, [17,18] showed similar results using even longer cues of 14–16 s.

The extended cues were more recognizable in two aspects: temporal sensitivity and heat flow sensitivity. Human thermal perception enables noticing a temporal difference of several seconds, but not fragments of a second, and the heat energy difference between a step and a ramp of a similar duration (the area under the temperature—time curve). This difference is key for designing thermal cues based on the concept of sequences of short pulses. The inferiority actually allows for more flexibility on the one hand and more reliability on the other.

### 5.4. Location in Sequence Impacts the Thermal Perception for the Individual Pulse

TECs create a heat flux, causing a temperature difference between the two sides that is proportional to the input current. Naturally, this temperature difference does not appear instantaneously, but rather develops in a parabolic-type curve, starting with a steep quasi-linear slope and then converging to the designated temperature that corresponds to the given stimulus. (It should be noted that the full temperature difference between the hot side and cold side of the TEC serving as our thermal display was actually approximately double the temperature change displayed, because the hot side did not remain stable when the stimulus was applied, and the dynamic thermal response to the stimulus was evenly divided between the sides. The hot side was monitored and investigated, but not shown in this report). The time coefficients of the temperature curve depend on the thermal characteristics of the components involved and the magnitude of the stimulus. In our case, the process would last several seconds if the stimuli would extend for that long, but since they continued for only 250 ms, the responses appeared linear for all practical purposes, with an ROC proportional to the current and a total temperature drop dictated by the pulse duration. In response to the termination of an electrical stimulus, a thermal recovery process begins, in which the system returns towards the baseline. Due to the shortness of the intervals between the pulses, the partial recoveries following the first two pulses of a trio were very limited. Consequently, the sequential pulses were applied at a non-steady state, as the thermal display temperature and skin temperature remained close to the resultant respective temperatures caused by the previous pulse. Hence, the location of a given pulse in the sequence dictated the initial conditions at which it was applied.

Correspondingly, two findings show that the location in the sequence impacted the thermal response to the individual pulse. One was psychophysical: an increased intensity sensed for a given stimulus magnitude. The other was physical: a reduced relative temperature drop caused by a given pulse. (The term ‘relative’ temperature drop refers to the nominal temperature drop normalized by the stimulus current). This phenomenon was due to the pulse being imposed in the opposite direction of an existing heat flow of recovery from the previous pulse. This is why it increased for the third pulse, because recovery was intensified by the increased temperature drop from the natural baseline skin temperature. These two findings are contrary to each other, and nevertheless, the results show that, despite the reduced temperature drop, the thermal perception not only did not decrease, but also intensified. This indicates that this is a powerful psychophysical phenomenon worth examining. Observing the thermoreceptor firing (action potential discharge) rates vs. the temperature in the range of interest (for this discussion, 28–34 °C is sufficient) shows that, as the temperature decreased, not only did the firing rate increase for the coolness receptors relative to the warmth receptors (which is trivial, for that is the cause of the cool sensation), but the gap was one of exponential growth. This indicates that the sensitivity to changes in temperature increased as the temperature decreased within the range.

This discussion emphasizes the robustness of the pulse concept, since the varying intensity was well sensed in the case of the sequential pulses, for which the recognizability of the pattern was based on relative sensations, and it can be used to intelligently design patterns for improved recognizability. However, for absolute sensation, the variations are meaningless.

## 6. Conclusions

We have proven the feasibility of conveying information via thermal sensation using cues composed of different sequences of quick pulses. Predetermined patterns encoding abstract information were applied to the thermal display, haptically sensed by touch, and decoded. Several sequences of two or three cooling pulses with a total duration of up to 2.75 s and a low impact on skin temperature (up to 2 °C) were designed and tested, showing a high recognizability. This potential capability offers an alternative or complementary channel for various scenarios in which conventional channels, vision, hearing, and tactile sensing are not applicable or sufficient (e.g., enhancing communication capability for the deafblind and transferring messages in noisy or silent environments).

The main inherent drawbacks in the feasibility test are as follows:A limited number of cues were tested. An effective data transfer method requires a variety of symbols.The tested cues included solely coolness cues—adding warmth pulses poses several challenges: the technical aspect of creating the stimuli, determining the human sensitivity to warmth pulses, and combining warmth and coolness in one cue with appropriate synchronization due to different temporal responses [22].Few participants. A reliability test requires a statistical sample.

Future work: To capitalize on the feasibility study, a broad set of thermal cues designed in accordance with the results and insights detailed in this report will be tested for their reliability using a large group of participants. Twelve cues can be derived directly from the existing designs: six cooling cues that include one single pulse, two duos (tests #1 and #2), three trios (tests #4 through #6), and six matching warmth cues. We plan to evaluate a set of 22 cues and aim for recognizability with a 95% reliability.

Furthermore, to enrich the variety of the thermal cues and expand the potential domains of their applicability, research should focus on three major efforts:A better utilization of the space dimension, i.e., enlarging the contact area with the thermal displays, thereby improving sensibility, and more interestingly, varying the stimulation locations using wearable items (such as a watch, a bracelet, or an article of clothing formed from a smart textile) encouraged by recent reports showing a certain level of spatial sensitivity [13,27,28].Evaluating additional types of thermal cues, e.g., a continuous periodic fluctuation of pulses with varying encoded frequencies.Thermal cues combined with other tactile senses to form a full haptic interface, thereby leveraging the relative advantages of each sense and enhancing noticeability and recognizability.

## Figures and Tables

**Figure 1 bioengineering-10-01156-f001:**
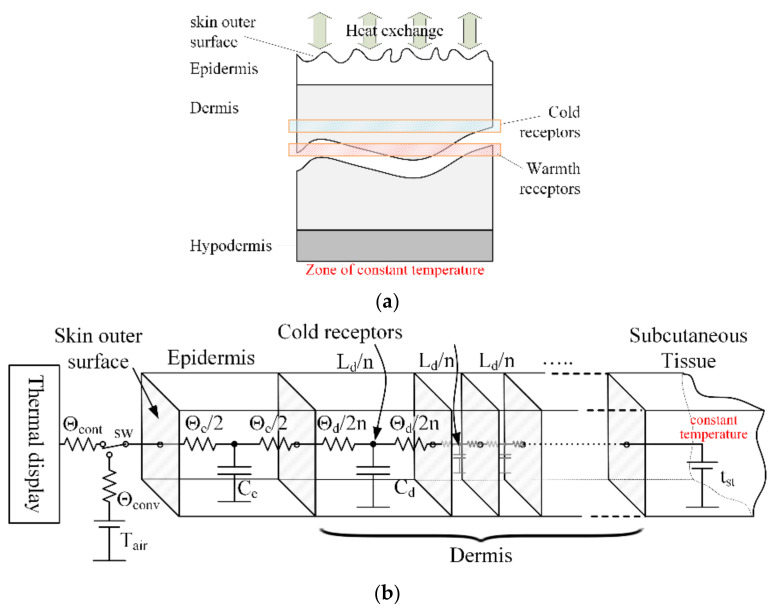
Schematic visualization of the human skin structure—(**a**) scheme of the fragment of skin based on [23,24], and (**b**) equivalent circuit model of the skin at the thermal contact zone based on data from [24,25]. The SW switch lets you choose between skin contact with the thermal display or ambient air. $\Theta_{cont}$, $\Theta_{conv}$, $\Theta_{e}$, and $\Theta_{d}$ are the thermal resistances of contact, convection, epidermis, and dermis, respectively, and $C_{e},\;C_{d}$ are the thermal capacities of the epidermal and dermal tissues, respectively. n is the number of layers, $L_{d}$ is the depth of the dermis, and $t_{st}$ is the subcutaneous temperature.

**Figure 2 bioengineering-10-01156-f002:**
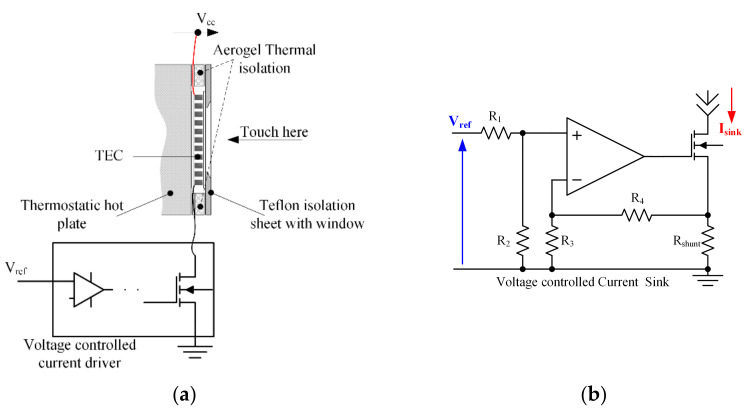
Scheme of the experimental setup. (**a**) Thermoelectric cooler (TEC), thermostatic plate, thermal insulation, current driver, and (**b**) current driver circuit, where $I_{sink}=V_{ref}\cdot R_{2}\cdot \left(R_{3}+R_{4}\right)\cdot {\left(R_{shunt}\cdot R_{3}\cdot \left(R_{1}+R_{2}\right)\right)}^{-1}$.

**Figure 3 bioengineering-10-01156-f003:**
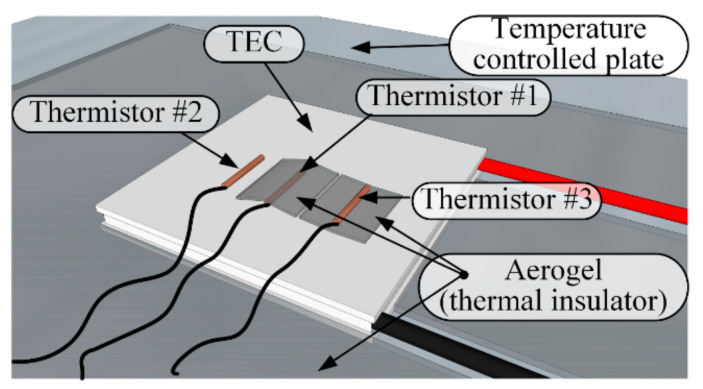
Scheme of the monitoring thermistor layout on the experimental thermal display. Thermistor **#1**—monitoring the temperature change of the thermal display when thermally isolated from the skin using a layer of Aerogel. Thermistor **#2**—monitoring display–skin interface temperature. Thermistor **#3**—Monitoring skin temperature at the contact zone by isolating it from the display.

**Figure 4 bioengineering-10-01156-f004:**
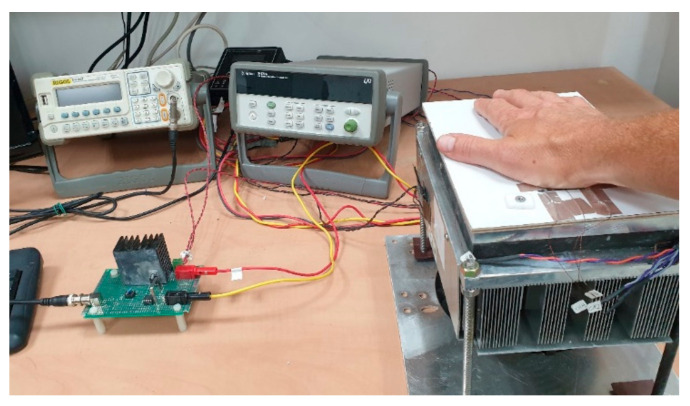
Laboratory layout—the picture shows (c.c.w. from right): The thermal display placed on the stabilized hot-plate and covered by the participant’s right hand, Agilent data acquisition, Rigol signals generator, and the current driver that receives the signal from the generator and transforms it into the input current for the TEC.

**Figure 5 bioengineering-10-01156-f005:**
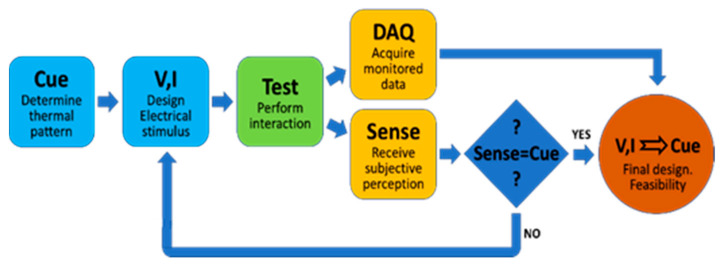
Illustration of the test procedure and data analysis—the trial-and-error process of correcting the electrical stimulus proceeds until clearly reaching the desired perception.

**Figure 7 bioengineering-10-01156-f007:**
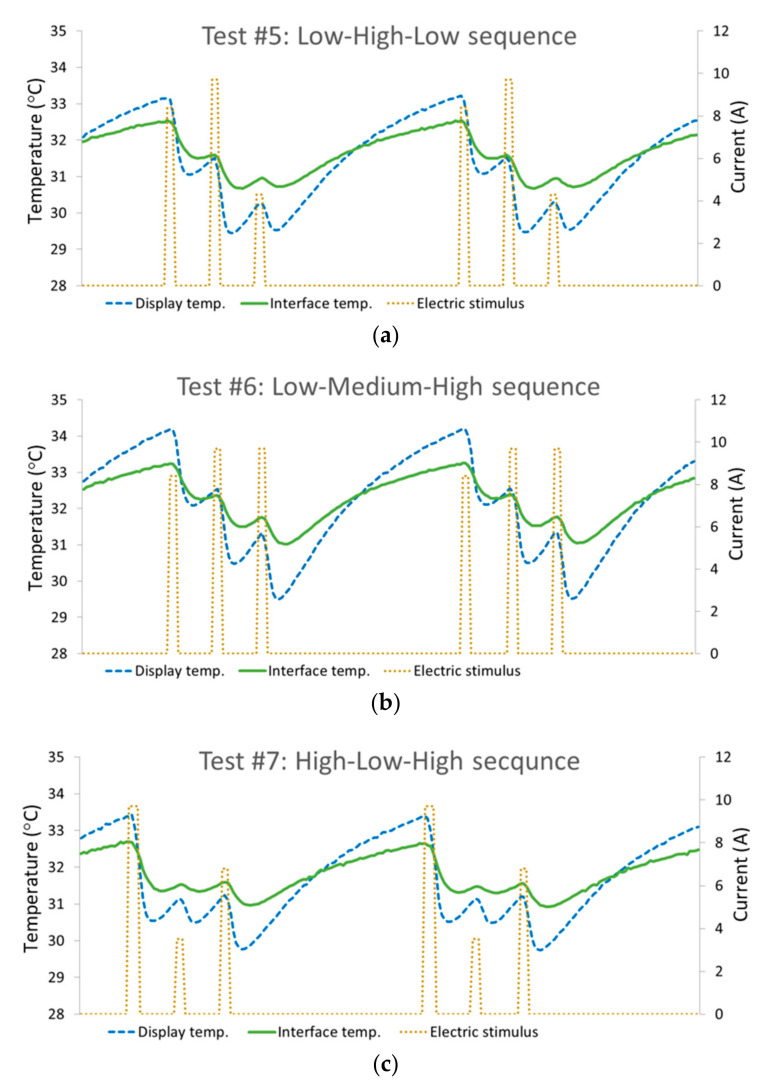
Temperature change vs. time in response to the Trio stimuli. The stimuli are presented as the electric current applied to the TEC (brown, dotted), the resultant thermal stimulus displayed by the TEC as monitored on the display with isolation from the skin (blue, dashed), and the cutaneous thermal response as monitored at the thermal display–skin interface (green, solid). The graphs randomly capture two of the many repetitions of the respective sequence applied periodically every 8 seconds. Three temperature drops are clearly viewed, each followed by a recovery attempt that is abruptly terminated by the consecutive pulse. The recovery curve after the third pulse is naturally longer and is stopped by the first pulse of the consecutive sequence, thereby defining the steady-state thermal equilibrium for the specific sequence. (**a**) Results from test #5—LHL trio. (**b**) Results from test #6—LMH trio. (**c**) Results from test #7—HLH trio.

**Table 1 bioengineering-10-01156-t001:** Description of tested patterns.

Test #	Sequence Definition	Icon *
1	Duo-Current—a sequence of two pulses with differentintensities (first Low and second High intensity), due tostimulus current	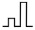
2	Duo-Duration—a sequence of two pulses with differentintensities (first Low and second High intensity), due tostimulus duration	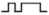
3	Pair of Duos—a pair of successive duos, one of identicalpulses, the other of different intensities	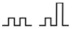
4	Pair of Duos—a pair of successive duos, both of differentintensities, one due to stimulus duration, the other dueto stimulus current	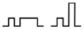
5	Trio LHL—a sequence of three pulses with differentintensities due to stimulus current—first Low intensity,second High, third Low	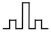
6	Trio LMH—a sequence of three pulses with differentintensities due to stimulus current—first Low intensity,second Medium, third High	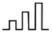
7	Trio HLH—a sequence of three pulses with differentintensities due to stimulus current—first High intensity,second Low, third High	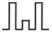

* The icons are intuitive graphic qualitative expressions of the patterns that refer to the corresponding thermal sensations. Note that upward pulses indicate a positive cooling sensation, counter to the direction of the temperature change.

**Table 2 bioengineering-10-01156-t002:** Desired thermal sensation vs. corresponding electrical stimuli and skin response.

Test #	ThermalSensation(Icon)	StimulusDesign(A)	Thermal Response—Skin Temperature Change (°C)
**5**	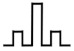	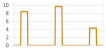	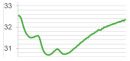
**6**	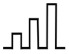	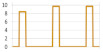	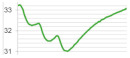
**7**	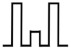	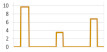	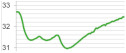

Remarks regarding Table 2: (a) Clarifying statement: For didactic purposes, it was decided to present the icons with the upward direction indicating a positive cooling sensation, with compatibility to the positive stimuli indicating a positive current applied to the TEC (causing a temperature decrease on the side serving as the thermal display), and counter to the direction of the skin temperature change. (b) Stimuli temporal parameters: All stimuli included three 250 ms pulses (except the first pulse of test #7 that was 300 ms) with intervals of 1 s. (c) The stimulus current should be referred to as a relative value since the absolute value derives from the specific test setup. (d) The presented thermal response captures the pulses and the recovery to baseline temperature.

## Data Availability

The data presented in this study are available on request from the corresponding author.

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
