# Peer review of "Thermal Cues Composed of Sequences of Pulses for Transferring Data via a Haptic Thermal Interface"

_bioengineering, 2023, doi:10.3390/bioengineering10101156_

Round 1

Reviewer 1 Report (Previous Reviewer 1)

The authors have addressed my concerns. This manuscript looks better now.

Reviewer 2 Report (New Reviewer)

The revised version is satisfied. This work could be accepted.

Minor editing of English language required.

This manuscript is a resubmission of an earlier submission. The following is a list of the peer review reports and author responses from that submission.

Round 1

Reviewer 1 Report

This paper presents a preliminary study of a haptic thermal interface for pulse coded data transferring. This is an interesting topic. However, this manuscript should be shortened and simplified.

1. It is better to provide more examples of potential applications and the advantages of using haptic thermal cues to transfer data over other cues in those applications to better show the motivations of this study.

2. The Background section is too long. It is better to limit the content of this section to the scientific facts most relevant to the design of haptic thermal interfaces without going into too much tedious analysis.

3. The main contribution of this paper is to demonstrate the feasibility of using the designed haptic thermal interface to transfer encoded data. You may show the comparison between the final optimized stimuli parameters and other configurations, but the detailed intermediate results in the optimization process of this study are not recommended to be shown too much in this paper.

Reviewer 2 Report

This study presented proof of the feasibility of transmitting information using haptic thermal signals. The main value added by the manuscript remains unclear, and the result is still limited. Nevertheless, the following specific questions should be answered and added to the manuscript to improve its clarity. There are some comments listed below:

1.     The introduction should be shortened and rewritten to clarify the study's purpose.

2.     It isn't obvious for readers to understand the experiment steps. There is some confusion why the author used seven different sequences in Table 1. In addition, there is only experimental content written in the text, and readers may be tired when reading it. I suggest deleting some content; it is better to be concise and comprehensive.

3.     The author should point out the microcontroller used in the paper, including the circuit design, dataset, and source code in the supplement section.

4.     What is the meaning of using a thermoelectric cooler (TEC) and a thermistor to measure the skin temperature? Does the room temperature affect the system? In Figure 6, the results in a and b are corresponding. That doesn't sound quite convincing. What is the function of electric stimulus in this case? Also, the same way happened in Figure 7 a, b, and c.

5.     Please add a table to compare this work with others to depict positive/negative points.

6.     The conclusion should be rewritten to clarify some drawbacks of this study, future work scope, and quantitative research findings.

N/A